# Empagliflozin Ameliorates Bleomycin-Induced Pulmonary Fibrosis in Rats by Modulating Sesn2/AMPK/Nrf2 Signaling and Targeting Ferroptosis and Autophagy

**DOI:** 10.3390/ijms24119481

**Published:** 2023-05-30

**Authors:** Hemat El-Sayed El-Horany, Marwa Mohamed Atef, Muhammad Tarek Abdel Ghafar, Mohamed. H. Fouda, Nahla Anas Nasef, Islam Ibrahim Hegab, Duaa S. Helal, Walaa Elseady, Yasser Mostafa Hafez, Rasha Youssef Hagag, Monira Abdelmoaty Seleem, Mai Mahmoud Saleh, Doaa A. Radwan, Amal Ezzat Abd El-Lateef, Rania Nagi Abd-Ellatif

**Affiliations:** 1Medical Biochemistry Department, Faculty of Medicine, Tanta University, Tanta 31511, Egypt; hemat.elhorany@med.tanta.edu.eg (H.E.-S.E.-H.);; 2Department of Biochemistry, College of Medicine, Ha’il University, Hail 81411, Saudi Arabia; 3Clinical Pathology Department, Faculty of Medicine, Tanta University, Tanta 31511, Egypt; 4Physiology Department, Faculty of Medicine, Tanta University, Tanta 31511, Egypt; 5Department of Bio-Physiology, Ibn Sina National College for Medical Studies, Jeddah 22421, Saudi Arabia; 6Pathology Department, Faculty of Medicine, Tanta University, Tanta 31511, Egypt; 7Anatomy and Embryology Department, Faculty of Medicine, Tanta University, Tanta 31511, Egypt; 8Internal Medicine Department, Faculty of Medicine, Tanta University, Tanta 31511, Egypt; 9Medical Pharmacology Department, Faculty of Medicine, Tanta University, Tanta 31511, Egypt; 10Chest Diseases Department, Faculty of Medicine, Tanta University, Tanta 31511, Egypt

**Keywords:** autophagy, bleomycin, empagliflozin, endoplasmic reticulum stress, ferroptosis, pulmonary fibrosis

## Abstract

Pulmonary fibrosis (PF) is a life-threatening disorder that severely disrupts normal lung architecture and function, resulting in severe respiratory failure and death. It has no definite treatment. Empagliflozin (EMPA), a sodium-glucose cotransporter 2 (SGLT2) inhibitor, has protective potential in PF. However, the mechanisms underlying these effects require further elucidation. Therefore, this study aimed to evaluate the ameliorative effect of EMPA against bleomycin (BLM)-induced PF and the potential mechanisms. Twenty-four male Wister rats were randomly divided into four groups: control, BLM treated, EMPA treated, and EMPA+BLM treated. EMPA significantly improved the histopathological injuries illustrated by both hematoxylin and eosin and Masson’s trichrome-stained lung tissue sections, as confirmed by electron microscopic examination. It significantly reduced the lung index, hydroxyproline content, and transforming growth factor β1 levels in the BLM rat model. It had an anti-inflammatory effect, as evidenced by a decrease in the inflammatory cytokines’ tumor necrosis factor alpha and high mobility group box 1, inflammatory cell infiltration into the bronchoalveolar lavage fluid, and the CD68 immunoreaction. Furthermore, EMPA mitigated oxidative stress, DNA fragmentation, ferroptosis, and endoplasmic reticulum stress, as evidenced by the up-regulation of nuclear factor erythroid 2-related factor expression, heme oxygenase-1 activity, glutathione peroxidase 4 levels, and a decrease in C/EBP homologous protein levels. This protective potential could be explained on the basis of autophagy induction via up-regulating lung *sestrin2* expression and the LC3 II immunoreaction observed in this study. Our findings indicated that EMPA protected against BLM-induced PF-associated cellular stress by enhancing autophagy and modulating sestrin2/adenosine monophosphate-activated protein kinase/nuclear factor erythroid 2-related factor 2/heme oxygenase 1 signaling.

## 1. Introduction

Pulmonary fibrosis (PF) is a chronic, progressive, age-related interstitial lung disease with a high morbidity and mortality rate. Its pathological features include excessive collagen deposition, enlarged interstitial spaces between alveoli, thickened alveolar walls, and inflammatory cell infiltration, resulting in widespread scarring, decreased lung compliance, and even lung failure [1]. There is currently no treatment, and animal models of PF are still crucial tools for investigating its etiopathogenesis. Although the etiology of PF is complex and variable, the accepted hypothesis is that it is caused by abnormal wound healing of alveolar epithelial cells in response to repeated injury stimulation [2]. Bleomycin (BLM)-induced PF is the most widely accepted model for developing new anti-fibrosis strategies and evaluating drug efficacy due to its low cost, ease of induction, and good reproducibility [3].

Ferroptosis is a recently described nonapoptotic form of programmed cell death that differs mechanistically and phenotypically from other well-known types of regulated cell death [4]. Biochemically, it is characterized by excessive reactive oxygen species (ROS) as a result of increased iron metabolism and lipid peroxidation, with no involvement of caspase or Bax activation. Morphologically, ferroptotic cells have smaller mitochondria, condensed mitochondrial membrane densities, vanishing mitochondrial cristae, and a ruptured outer mitochondrial membrane [5]. Glutathione peroxidase 4 (GPX4) is a key regulator of ferroptosis. It reduces lipid hydroperoxide production during lipid peroxidation, which is involved in oxidative damage to cellular membrane structure [6]. Recent studies have documented the role of ferroptosis in the pathophysiology of several conditions, including neurodegeneration, ischemic organ injury, atherosclerosis, and cancer [7]. Notably, recent studies have demonstrated that inhibiting ferroptosis can alleviate experimentally induced lung injuries such as lipopolysaccharide-induced acute lung injury [8] and radiation-induced lung fibrosis [9]. However, the role of ferroptosis in BLM-induced PF remains unknown. Therefore, we hypothesize that ferroptosis is a possible underlying mechanism.

Nuclear factor erythroid 2-related factor 2 (Nrf2) plays an important bidirectional role in cancer prevention and progression. As a well-known cytoprotective factor, it plays an important role in protecting cells from oxidative damage caused by increased reactive oxygen species (ROS) levels. When activated by ROS, Nrf2 induces the expression of antioxidant enzymes by binding to the promoters of target genes containing antioxidant response elements, thereby contributing to the synthesis and renewal of glutathione (GSH), a major GPX4 cofactor, and detoxifying molecules [10,11]. Therefore, Nrf2 activation plays a protective role under physiological conditions. However, Nrf2 activation in cancer cells promotes cancer progression and metastasis, as well as chemoresistance by inactivating drug-mediated oxidative stress and protecting cancer cells from drug-induced cell death. Selective Nrf2 modulators can be used as adjuvant therapy after conventional chemotherapy, targeted therapy, and immunotherapy [11]. Targeting the Nrf2 pathway in cancer cells could pave the way for the development of more effective anticancer drugs or improve the efficacy of existing drugs by reducing drug resistance [12]. Furthermore, Nrf2 expression can be reactivated by natural or synthetic compounds via downregulating histone deacetylases and DNA methyltransferases, inhibiting *Nrf2* promoter methylation and protecting normal cells from ROS damage and tumorigenesis [13]. Growing evidence suggests a link between Nrf2 and ferroptosis. In addition, many genes that inhibit iron overload and lipid peroxidation have been identified as *Nrf2* target genes [14,15]. Adenosine monophosphate-activated protein kinase (AMPK) is a key bioenergetic sensor and metabolic regulator. Recent studies indicated that AMPK activation could enhance Nrf2 nuclear translocation [16].

Sestrin2 (Sesn2) is a stress-inducible protein that is strongly up-regulated by various stressors. It has protective effects on various physiological and pathological states, mainly by regulating oxidative stress, endoplasmic reticulum (ER) stress, autophagy, and inflammation [17]. It has recently been shown to efficiently affect the pathological process of ferroptosis [18]. In addition, emerging evidence suggests that Sesn2 plays a pivotal role in fibrosis progression in many organs, including the liver and heart [19,20]. Furthermore, Sesn2 induction has antioxidant and anti-inflammatory effects in alleviating cigarette-smoke-induced pulmonary injury [21].

Empagliflozin (EMPA) is a sodium-glucose cotransporter 2 (SGLT2) inhibitor that was developed as a glucose-lowering agent for type 2 diabetes mellitus. It controls glucose levels by lowering glucose uptake by renal proximal tubular cells without causing hypoglycemia, thereby improving insulin resistance [22]. Beyond its antidiabetic effect, recent research has shown that EMPA has beneficial effects on the cardiovascular, renal, hepatic, and pulmonary systems, which may be due to its ability to reduce inflammation, oxidative stress, fibrosis, apoptosis, and mitochondrial dysfunction [23,24,25,26]. Although EMPA has recently been shown to ameliorate BLM-induced PF by activating Nrf2 [27], more research is required to elucidate the mechanisms underlying EMPA’s lung protective effects against BLM. Therefore, the current study aimed to investigate EMPA’s regulatory effect on the Sesn2/AMPK/Nrf2/heme oxygenase 1 (HO-1) signaling pathway as well as its modulatory role in ferroptosis, inflammation, ER stress, and autophagy in the context of BLM-induced PF.

## 2. Results

### 2.1. Effects of EMPA on BLM-Induced Histopathological Alterations

Histopathological examination of hematoxylin and eosin (H and E)-stained sections from both the control and EMPA-treated groups revealed normal lung tissues with alveolar sacs and alveoli separated by thin interalveolar septa. They exhibited type I pneumocytes with flat nuclei, and a small number of type II pneumocytes with centrally rounded vesicular nuclei lined the alveoli (Figure 1A,C). However, lung tissue sections from the BLM-treated group revealed disrupted lung architecture with marked thickening of interalveolar septa and significant mononuclear cellular infiltrations. Many of the alveolar epithelial lining cells showed vacuolar cytoplasmic necrosis and pyknotic nuclei. Some collapsed alveoli and severe interstitial hemorrhage could be seen (Figure 1B). However, the normal lung tissue, alveolar sacs, and alveoli were all preserved in the EMPA+BLM-treated group, except for mild thickening of the alveolar septa and mild interstitial hemorrhage (Figure 1D). In addition, the histopathological score significantly increased in the BLM-treated group as compared to the control groups. In contrast, EMPA improved the histopathological score more effectively in group IV (Figure 1E). The scoring of lung fibrosis in the BLM-treated group revealed significantly more fibrotic changes in the interalveolar septa, blood vessel walls, and bronchioles when compared to the control group. Furthermore, the EMPA+BLM-treated group showed a significant reduction in lung tissue fibrosis score when compared to the BLM-treated group (Figure 1F).

Furthermore, histopathological examination of Masson’s trichrome-stained lung tissue sections from the control and EMPA-treated groups revealed negligible collagen fibers in the alveolar walls (Figure 2A,C). However, the BLM-treated group revealed an increased amount of collagen fibers (Figure 2B). Notably, EMPA administration to BLM-treated rats significantly reduced the amount of collagen fibers (Figure 2D). The morphometric measurements of Masson trichrome-stained sections revealed that the BLM-treated group had a significant increase in the mean area percentage of Masson trichrome when compared to the control and EMPA-treated groups, while the EMPA+BLM-treated group had a considerable decrease when compared to the BLM-treated group (Figure 2E).

### 2.2. Effects of EMPA on BLM-Induced Inflammatory Status

The examination of CD68 immunohistochemical (IHC)-stained sections revealed a few CD68-positive brownish macrophages that were distributed in alveolar walls in both the control and EMPA-treated groups (Figure 3A,C). Nevertheless, the BLM-treated group revealed numerous cells with a very strong CD68 immunoreaction (Figure 3B). On the other hand, the EMPA + BLM-treated group showed numerous CD68-positive macrophages with a strong positive immunoreaction (Figure 3D). Morphometric analysis of the mean percentage of positive CD68 immunoreactions revealed a highly significant increase in the BLM-treated group when compared to the control group. However, the EMPA+BLM-treated group revealed a significant decrease when compared to the BLM-treated group (Figure 3E).

Likewise, the total leucocytic count (TLC) (Figure 3F), as well as the percentages of neutrophils and lymphocytes in the bronchoalveolar lavage fluid (BALF) (Figure 3G,H), were significantly higher in the BLM-treated group than in the control and EMPA-treated groups, while the macrophage percentage was significantly lower (Figure 3I). Meanwhile, concomitant EMPA administration to BLM-treated rats significantly restored the leucocytic parameters to near-control values (Figure 3). In the BLM-treated group, there were significant increases in both BALF tumor necrosis factor-alpha (TNF-α) and high mobility group box 1 (HMBG1) levels when compared to the control and EMPA-treated groups. EMPA, on the other hand, attenuated inflammatory responses, as evidenced by significant reductions in TNF-α and HMBG1 levels in BALF (Figure 3J,K). These findings suggested that EMPA could attenuate the BLM-induced inflammatory status.

### 2.3. Effects of EMPA Treatment on BLM-Induced Lung Fibrosis

When compared to the control and EMPA-treated groups, BLM treatment resulted in a significant increase in the lung index in the BLM-treated group. However, this increase was ameliorated by concomitant EMPA administration to the BLM-treated rats (Figure 4A). In the BLM-treated group, there were significant increases in the fibrotic markers transforming growth factor beta 1 (TGF-β1) and hydroxyproline levels in lung tissues when compared to the control and EMPA-treated groups. EMPA, on the other hand, attenuated fibrotic responses, as evidenced by significant reductions in TGF-β1 and hydroxyproline levels in lung tissues (Figure 4B,C). These findings suggested that EMPA could attenuate BLM-induced lung fibrosis.

### 2.4. Effects of EMPA on Redox Status, Ferroptosis, and ER Stress Status

BLM treatment significantly increased malondialdehyde (MDA) and 4-hydroxynonenal (4-HNE) levels while reducing GSH and GPX4 levels and HO-1 activity in the BLM-treated group compared to the control and EMPA-treated groups. However, these changes were reversed by EMPA treatment (Figure 5A–E). Similarly, *Nrf2* relative expression level was found to be significantly down-regulated in the BLM-treated group compared to the control and EMPA-treated groups. However, concomitant EMPA administration to BLM-treated rats significantly upregulated *Nrf2* relative expression levels compared to those treated only with BLM (Figure 5F). Furthermore, the levels of C/EBP homologous protein (CHOP), a mediator of ER stress-induced apoptosis, were significantly increased after BLM treatment compared to the control and EMPA-treated groups. However, the coadministration of EMPA attenuated the BLM effect on CHOP levels (Figure 5G). These findings suggested that EMPA could attenuate the BLM-induced oxidative and ER stress and ferroptosis in the BLM rat model.

### 2.5. Effects of EMPA on Autophagy Regulation

The examination of LC3 II IHC-stained sections revealed many cells with strong positive LC3 II immunoreactions in their cytoplasm in both the control (Figure 6A) and EMPA-treated groups (Figure 6C). However, the BLM-treated group revealed a few cells with a very weak LC3 II immunoreaction (Figure 6B). Nonetheless, sections from rats cotreated with EMPA showed numerous LC3 II-positive cells with a strong positive immunoreaction (Figure 6D). Furthermore, morphometric analysis of the mean percentage of positive LC3 II immunoreaction revealed a highly significant decrease in the BLM-treated group when compared to the control group, whereas concomitant EMPA administration to BLM-treated rats revealed a significant increase when compared to the BLM-treated rats (Figure 6E). Likewise, *Sens2* relative expression levels were found to be significantly downregulated in the BLM-treated group compared to the control and EMPA-treated groups. However, concomitant EMPA administration to BLM-treated rats significantly up-regulated *Sens2* relative expression levels compared to those treated only with BLM (Figure 6F). Indeed, the p-AMPK levels in lung tissues were significantly lower in the BLM-treated group than in the control group. Meanwhile, concomitant EMPA administration to BLM-treated rats significantly increased lung p-AMPK levels compared to those treated only with BLM (Figure 6G). These findings suggested that EMPA could promote autophagy in BLM rat model.

### 2.6. Effects of EMPA on DNA Damage

DNA samples from both the control and EMPA-treated groups showed normal, distinct bands (3162 and 3061 bp, respectively) in the DNA ladder assay. The BLM-treated group, in contrast, displayed an apoptotic fragmented DNA pattern. When EMPA was administered concurrently with BLM, DNA fragmentation was reduced, protecting against BLM’s adverse effects (Figure 7).

### 2.7. Effects of EMPA Treatment on the Electron Microscopic Alterations in the Lung Specimens

Electron microscopic examination of the ultrathin sections of the lung specimens revealed normal ultrastructure features of the lung. Type I pneumocytes appeared flattened in shape, with clearly demonstrated flat nuclei filling a large part of the cytoplasm. Type II pneumocytes were cuboidal in shape with euchromatic, rounded nuclei and scattered microvilli. The most distinguishing feature of these cells was the presence of numerous cytoplasmic lamellar bodies. Alveolar macrophages were detected with large, indented nuclei and prominent nucleoli, a large number of lysosomes in their cytoplasm, and multiple pseudopodia on their surface in both the control and EMPA-treated groups (Figure 8A,D). However, ultrathin sections from the BLM-treated group revealed marked perturbations of type II pneumocytes, with irregularly shrunken nuclei, empty lamellar bodies, disturbed mitochondria, and detached microvilli in the alveolar space, as well as thick collagen fibrils and edematous fluid in the interstitial tissue (Figure 8B,C). Concomitant EMPA and BLM treatment improved lung ultrastructure, resulting in type II pneumocytes with large euchromatic nuclei. Few mitochondria remained disrupted, and the microvilli were mostly intact. Some lamellar bodies were partially empty, while others had lung surfactant strands (Figure 8E).

## 3. Discussion

As novel evidence, this study clearly demonstrates the protective role of EMPA in BLM-induced PF in experimental animals by ameliorating oxidative stress, ferroptosis, inflammation, ER stress, and fibrosis, as well as improving histopathological changes and autophagic activity in lung tissues. This protective potential may be related to the activation of the Sesn2/AMPK/Nrf2/HO-1 signaling pathway.

The BLM model is one of several PF animal models that is widely used due to its superior characteristics. According to the current study, BLM caused pulmonary inflammation and fibrosis, as evidenced by lung histopathological findings, significant elevations in various inflammatory mediators such as TNF-α, HMG1, and BALF TLCs, and a very strong CD68 immunoreaction revealed by immunostaining, as well as the lung index, TGF-β1 levels, and the primary building block for collagen content, hydroxyproline, which is consistent with previous findings [28]. These pathological alterations were markedly reversed by EMPA, consistent with Kabel et al. [27], and supported by EMPA’s anti-inflammatory and anti-fibrotic potential demonstrated in different models of organ fibrosis in the absence of diabetes [29,30].

A growing body of evidence suggests that ROS plays an important role in the pathogenesis of PF. High levels of ROS promote TGF-β1-induced fibrosis and the fibroblast-to-myofibroblast transition, as well as inflammatory cytokine secretion [28]. Furthermore, high ROS levels induce lipid peroxidation, which triggers ferroptosis and subsequent fibrosis-related diseases such as PF [9,31,32]. Li et al. demonstrated the effectiveness of ferroptosis inhibition on radiation-induced lung fibrosis by downregulating proinflammatory cytokines mediated by Nrf2 pathway activation [9]. Therefore, we hypothesized that inhibiting ferroptosis could be an effective strategy for ameliorating BLM-induced PF.

GPX4 is a key regulator of ferroptosis. It has been documented that GPX4 overexpression inhibits ferroptosis, whereas *GPX4* deletion results in severe pathological phenotypes related to ferroptosis [33]. The current findings demonstrated the main characteristics of ferroptosis, including MDA burden, 4-HNE accumulation, reduced GPX4 levels, and GSH depletion. The decreased GPX4 levels could be explained by GSH depletion, ROS accumulation, and lipid peroxidation, exceeding GPX4 clearance ability [34].

Intriguingly, there was a significant decrease in MDA and 4-HNE levels in the EMPA + BLM-treated group, indicating alleviated oxidative stress and lipid peroxidation, accompanied by significantly increased GSH and GPX4 levels. These findings could imply that EMPA exhibited remarkable free radical scavenging activities, reversed the reduction in antioxidant enzyme activity, and regulated ferroptosis. In line with this, in vivo and in vitro studies demonstrated that EMPA could reduce ROS generation in mitochondria and cytoplasm [35,36]. According to previous studies, EMPA may have the ability to suppress ferroptosis in different animal models [37,38]. To the best of our knowledge, this represents the first study that demonstrated the beneficial effects of EMPA treatment on the regulation of ferroptosis in PF, and we inferred that the reduction in GSH content and inhibition of system Xc- might be the targets of EMPA to act upon, as previously reported [39].

Nrf2 primarily regulates ferroptosis by directly affecting the expression and function of GPX4. In addition to regulating intracellular iron concentration via the Nrf2/HO-1 axis, the Nrf2 signaling pathway promotes cystine/glutamate reverse transport system Xc- and *GPX4* expression [14]. In the meantime, knocking down *Nrf2* and *Nrf2*-target genes enhanced ferroptosis-mediated cell death [39]. Previous research has shown that upregulating *Nrf2* expression and its dependent antioxidant factors, including HO-1, can mitigate oxidative stress-induced pulmonary inflammation and fibrosis in the BLM-induced PF model [40]. In the same context, the present findings showed that BLM reduced *Nrf2* expression levels and HO-1 activity, which were significantly associated with GPX4 and GSH levels, supporting the novel role of Nrf2 in PF pathogenesis by regulating ferroptosis [40]. Furthermore, these effects were reversed by EMPA treatment, indicating that EMPA’s antioxidant effects are mainly through activating Nrf2 signaling, as evidenced in different models and tissues [25,41,42]. Therefore, we hypothesized that ferroptosis may be one of the pathogenetic mechanisms in BLM-induced PF, and EMPA treatment may regulate ferroptosis to ameliorate PF partly by reducing lipid peroxidation, suppressing oxidative stress, enhancing GSH production, and increasing GPX4 levels via the Nrf2/HO-1 signaling pathway. However, the mechanism by which EMPA activates *Nrf2* expression requires further investigations to uncover the upstream regulatory factors.

Autophagy is a fundamental cytoprotective mechanism that maintains cellular homeostasis. It degrades and recycles damaged cytoplasmic constituents in response to cellular stress [43]. There is mounting evidence supporting the pathogenic role of impaired autophagy in PF, as well as the ameliorative effects of autophagy induction on disease progression [44,45]. Sesn2, an important sestrin member, is involved in the positive regulation of autophagy [46]. Evidence suggests that Sesn2 is involved in oxidative-stress-related respiratory diseases [47]. It has also been shown to inhibit fibrogenic and inflammatory cytokine production [48]. However, its regulatory role in autophagy in PF remains unknown.

The current findings revealed that BLM significantly reduced *Sesn2* expression as well as p-AMPK levels, which is a downstream target modulated by Sesn2 and one of the classic core molecular regulators of autophagy. These findings were associated with a significant decrease in LC3-II levels revealed by immunostaining, indicating a blockade of autophagic flux, which is consistent with Sosulski et al. [44]. In addition, Yang et al. found that *Sesn2* was downregulated in TGF-β-treated cells and BLM-treated mice, whereas its up-regulated expression induced autophagy and alleviated PF [49]. Furthermore, Shi et al. reported that activating AMPK could ameliorate BLM-induced lung fibrosis [50]. In this study, reduced *Sesn2* expression could be attributed to reduced *Nrf2* expression, which is an essential transcription factor for *Sesn2* expression in response to stress [17]. In contrast, Zehender et al. revealed that autophagy activation was required for fibroblast activation in dermal and pulmonary fibrosis [51]. Sesn2 was also found to increase dramatically in LPS-induced acute lung injury [52]. These contradictory results regarding the paradoxical status of *Sesn2* expression may be related to the different periods of lung injury, time window, oxidant stress level, and cell phenotype.

Several studies have reported the role of EMPA in inducing autophagy, which is consistent with our findings [53,54]. However, the impact of EMPA on *Sesn2* expression has yet to be investigated. In this study, we found that EMPA significantly increased *Sesn2* expression and AMPK phosphorylation, as well as increased LC3-II immunostaining, indicating increased autophagy activity. In the same vein, Sun et al. reported that EMPA improved obesity-related cardiac dysfunction via promoting Sesn2-mediated AMPK-mTOR signaling and Sesn2-induced Nrf2 activation by enhancing the autophagic degradation of Keap1 [55].

Furthermore, given the role of AMPK in Nrf2 phosphorylation and nuclear accumulation [56], we could speculate that EMPA-induced *Nrf2* expression and ferroptosis regulation depend on enhancing the Sesn2/AMPK signaling pathway. This finding was consistent with previous findings by Lu et al. [37]. Furthermore, EMPA improved autophagy and up-regulated *Sesn2* expression by enhancing the SIRT1 signaling pathway [57,58].

Given the close relationship between ER stress and autophagy, as well as the role of ER in the development of lung fibrosis [59], we investigated whether EMPA could protect the lungs via an ER-stress-associated mechanism. CHOP is a transcription factor that is used as a marker for ER stress and cell survival [60]. Our findings revealed that BLM significantly increased CHOP levels in lung tissues, which were significantly ameliorated by EMPA treatment, as previously reported [45,61]. Hence, we could postulate that the potential effect of EMPA on mitigating ferroptosis, inflammation, and ER stress in our model was due to its positive impact on *Sesn2* expression, as previously reported [18].

Notably, BLM causes single- and double-stranded DNA breaks, which are linked to accelerated DNA fragmentation [62]. In the present study, DNA fragmentation was assessed via the DNA ladder assay method. BLM treatment caused significant DNA band fragmentation. This finding was consistent with previous findings by Hu et al. [63]. Meanwhile, EMPA coadministration reduced DNA fragmentation. Similarly, Min et al. and Ekici et al. showed that EMPA protected against DNA damage in other animal models [64,65].

There are some limitations to this study. First, the precise molecular mechanism underlying EMPA-mediated regulation of autophagy and ferroptosis has yet to be evaluated. Second, the mechanism of action of EMPA needs further investigation by knocking down or overexpressing the related proteins. Third, whether co-treatment with EMPA affects the tumor-killing potency of BLM, as well as whether EMPA is a safe drug for patients receiving BLM, remains to be determined.

## 4. Materials and Methods

### 4.1. Chemicals

Bleomycin (15 U/vial) was purchased from Celon Laboratories Pvt. Ltd., Hyderabad, India (NDC 61703-332-18). Empagliflozin (Jardiance) was obtained from Boehringer Ingelheim, Ingelheim am Rhein, Germany (NDC 0597-0152-30), and all other chemicals were obtained from Sigma Aldrich Chemical Co., Ltd. (St. Louis, MO, USA).

### 4.2. Experimental Design

This experiment was conducted on 24 male *Wistar* rats obtained from the animal house of Tanta University, Egypt, and weighing 160 to 200 g. Throughout the experiment, rats were kept at 22 ± 2 °C with 12 h dark/light cycles and access to chow and water ad libitum. After a one-week acclimatization period, rats were randomly divided into four experimental groups of six rats each (Figure 9). Group I (the control group) received saline intraperitoneally in the same manner as BLM injections, as well as 1% carboxymethyl cellulose (CMC) orally in the same manner as EMPA. Group II (the BLM-treated group) received BLM (15 mg/kg) intraperitoneally three times per week for four successive weeks in order to induce pulmonary fibrosis [66,67]. Group III (the EMPA-treated group) received EMPA dissolved in 1% CMC orally via oral gavage at a dose of 10 mg/kg/day throughout the experimental period [27,68]. Group IV (the combined EMPA and BLM-treated group) received EMPA (10 mg/kg) orally via oral gavage seven days before BLM administration and continued for four weeks after BLM injection (Figure 9) [67].

At the end of the experiment, the overnight fasted rats were weighed, anesthetized with thiopental sodium (20 mg/kg), and sacrificed. After opening the thoracic cavity, BALF was collected as described below and stored at −80 °C. Lungs were harvested, soaked in ice-cold saline, and weighed, and the lung index was calculated as lung wet weight (mg)/body weight (g). The left lung was then fixed in a 10% formaldehyde solution for histopathological and IHC examination. The right lung was stored at −80 °C until it was used for tissue homogenate preparation, nuclear extraction, microsomal separation, and RNA extraction. Total protein concentrations in lung tissue homogenates, nuclear fractions, and microsomes were determined using Lowry’s method and bovine serum albumin as a standard [69].

The study protocol was approved by the Ethics Committee of the Faculty of Medicine, Tanta University, Egypt (approval no. 35760/9/22) and followed the National Institutes of Health guide for the care and use of laboratory animals (NIH Publications No. 85 –23, revised in 1996).

### 4.3. BALF Collection and Measurement of Inflammatory Markers

The thoracic cavity was opened, and the tracheas were exposed and cannulated with a blunt needle attached to a syringe. Bronchoalveolar lavage was performed three times with 0.8 mL of sterile phosphate-buffered saline (PBS). The fluid was then centrifuged at 2000 rpm for 10 min at 4 °C. Cell pellets were collected and resuspended in 500 µL of sterile PBS for total and differential cell counts [70]. BALF supernatant was collected and used to estimate TNF-α and HMGB 1 levels using enzyme-linked immunosorbent assay (ELISA) kits (RayBiotech Inc., Peachtree Corners, GA, USA, and Sunred biological technology, Shanghai, China, respectively).

### 4.4. Determination of Hydroxyproline Content and TGF-β1 in Lung Tissues

Hydroxyproline was estimated as a biochemical index of fibrosis using a colorimetric assay, as previously described [71]. In brief, samples were hydrolyzed with 6 N HCl for 24 h at 120 °C, then treated with the buffer and chloramine T reagent for 20 min at room temperature before being treated with perchloric acid for 15 min at 60 °C. After cooling, the absorbance of the red complex was measured at 550 nm. TGF-β1 levels were measured in lung tissue homogenates using an ELISA kit (Uscn Life Science Inc., Wuhan, China) according to the manufacturer’s protocol.

### 4.5. Assessment of Lung Redox Status and Ferroptosis Markers

The MDA levels in lung tissue homogenates were determined using the thiobarbituric acid method, as previously described by Ohkawa et al. [72]. The 4-HNE levels in lung tissue homogenates were assessed using an ELISA kit (Fine Biotech, Wuhan, China). The GSH levels were measured colorimetrically using a commercial kit supplied by Biodiagnostic, Egypt, according to the manufacturer’s instructions. Microsomal separation was performed according to the method of Schenkman et al. [73]. The microsomal HO-1 activity was measured using a modified method of Tenhunen et al. [74], in which the amounts of NADPH consumed were calculated using the molar extinction coefficient (ε = 6.220 M^−1^cm^−1^). The GPX4 levels in lung tissue homogenates were assayed using an ELISA kit purchased from MyBioSource (San Diego, CA, USA) according to the manufacturer’s protocol.

### 4.6. Determination of Active p-AMPK and CHOP in Lung Tissue Homogenates

The p-AMPK levels in lung tissue homogenates were measured using an ELISA kit purchased from Glory Science Co., Ltd., Del Rio, TX, USA. Nuclear extraction of lung tissues was carried out using a Membrane, Nuclear and Cytoplasmic Protein Extraction Kit (Bio Basic Inc., Markham, ON, Canada), as previously described [75]. The nuclear extracts of lung tissues were then assayed for CHOP levels using an ELISA kit purchased from MyBioSource, San Diego, CA, USA.

### 4.7. Real-Time Polymerase Chain Reaction

Total RNA was isolated from rat lung tissues using the GeneJET RNA Purification Kit (Thermo Scientific; catalog no. K0731). The extracted RNA (5 μg) was reverse transcribed into cDNA using RevertAid H Minus Reverse Transcriptase (Thermo Scientific; catalog no. EP0451). *Sesn2* and *LC3-II* expressions were relatively quantified using the StepOnePlus real-time PCR system (Applied Biosystems, Waltham, MA, USA). The primers’ sequences were as follows: *Sesn2* (GenBank accession No. NM_001109358.2; F: 5′-GGCTGTGGGATACTTCCTGA-3′ and R: 5′-TTCAATGGGTCTCTGCTTGG-3′), *Nrf2* (GenBank accession No. NM_001399173.1; F: 5′-ATTGCTGTCCATCTCTGTCAG-3′ and R: 5′-GCTATTTTCCATTCCCGAGTTAC-3′), and the housekeeping gene (*GAPDH*) (GenBank accession No. NM_017008.4; F: 5′-CATGCCGCCTGGAGAAACCTGCCA-3′ and R: 5′-GGGCTCCCCAGGCCCCTC CTGT-3′). The cycle threshold (Ct) values for the target and housekeeping genes were determined, and relative gene expression was calculated using the 2^−∆∆Ct^ method [76].

### 4.8. DNA Fragmentation by the Ladder Assay

The DNA was extracted and fragmented as described by Verma et al. [70]. The lung tissues’ DNA was extracted using the GF-1 Tissue DNA Extraction Kit (Vivantis; Shah Alam, Selangor Darul Ehsan, Malaysia). The high-purity genomic DNA was then eluted in low-salt or water-based buffers. The DNA purity and concentration were then measured at 260 nm. Following that, a 1.5% agarose gel was loaded with 5 µL of each sample and 1 µL of loading dye (6× DNA Loading Dye, Fermentas International, Burlington, ON, Canada). The voltage was adjusted to 80 volts, and the time was adjusted to 45 min. After the run has been completed, the gel was placed on the gel documentation system (Analytic Jena, Biometra model: Biodoc analyzer) for visualization, imaging, and analysis.

### 4.9. Histopathologic and IHC Assays

Paraffin-embedded blocks were prepared from left lung tissue specimens. For histological examination of lung tissue injury, 5 µm thick sections were stained with H and E and Masson’s trichrome stain. For IHC staining, 5 μm thick sections were deparaffinized, rehydrated, and washed in PBS, and endogenous peroxidase activity was inhibited with 3% hydrogen peroxide. The sections were then incubated at 4 °C with the primary antibodies, rabbit monoclonal anti-LC3II (0.5 mg/mL; 1:400; catalog no. ab232940; Abcam, MA, USA), and macrophage CD68 (1 mg/mL; 1:400; Bio-Rad, Alfred Nobel Drive, Hercules, CA, USA). To detect immunoreactivity, the sections were treated with a biotinylated secondary antibody, streptavidin peroxidase conjugate, and diaminobenzidine chromogen. The sections were counterstained with Mayer’s hematoxylin and morphometrically analyzed using a Leica Qwin 500 C Image analyzer (Leica Imaging System Ltd., Cambridge, UK), with photomicrographs evaluated. At a magnification of 400×, ten randomly selected nonoverlapping fields from each slide were quantified for the mean alveolar septal thickness in H and E-stained sections, the mean area percentage of collagen fiber content in Masson trichrome-stained sections, and the mean percentage and color intensity of positive IHC reactions for LC3II and CD68 in diaminobenzidine-stained sections.

Histopathological scoring was performed. The interalveolar septa thickening, interstitial mononuclear cell infiltrations, alveolar spaces collapse, bronchiolar epithelial lining thickening, and blood vessel wall thickening and congestion were all graded on a 0–3 scale, with 0 denoting no changes, 1 denoting mild changes, 2 denoting moderate changes, and 3 denoting severe changes. Pulmonary fibrotic changes were graded using the following scale: 0 (normal lung) denotes no fibrosis; 1 (isolated alveolar septa with mild fibrosis); 2 (fibrotic changes in the alveolar septa with knot-like formation); 3 (contiguous fibrotic walls of the alveolar septa); 4 (single fibrotic mass); 5 (confluent fibrotic mass); 6 (large contiguous fibrotic masses); 7 (an air bubble); and 8 (fibrous obliteration) [77].

### 4.10. Electron Microscopy

The lung tissues were fixed in 2.5% glutaraldehyde buffer and then in osmium tetroxide. They were then washed and dehydrated with alcohol. They were inserted in epoxy resin to prepare blocks that were later used for cutting ultra-thin sections, which were stained with uranyl acetate and lead citrate. Photographs were captured by a JEOL 2100 transmission electron microscope (Tokyo, Japan) [78,79].

### 4.11. Statistical Analysis

All data were analyzed using the Statistical Package for the Social Sciences (SPSS) software, Version 20.0. For each group of six rats, the data were expressed as means and standard deviations. The one-way analysis of variance (ANOVA) followed by Tukey’s test was used to compare data among groups. A *p*-value of less than 0.05 was considered statistically significant.

## 5. Conclusions

The current study is the first to demonstrate that EMPA has a promising protective effect against BLM-induced PF in rats by enhancing autophagy and mitigating ferroptosis, inflammation, and ER stress via modulating the Sesn2/AMPK/Nrf2/HO-1 signaling pathway. However, further studies are warranted to confirm the protective effect of SGLT2 inhibitor therapy, such as EMPA, against pulmonary complications in patients receiving BLM and to determine the extent to which it can antagonize BLM’s therapeutic effect in such patients by promoting chemoresistance.

## Figures and Tables

**Figure 1 ijms-24-09481-f001:**
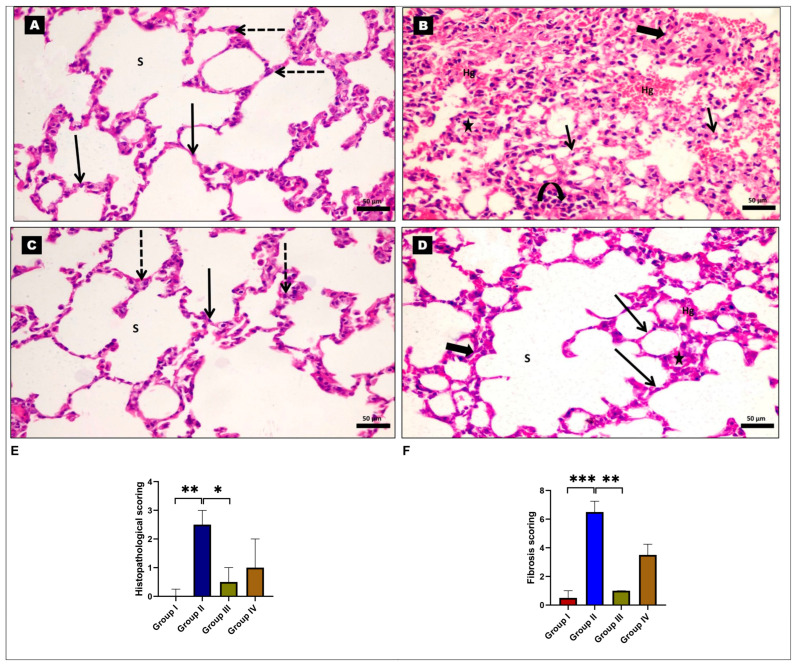
Representative photomicrographs of rat lung tissues stained with hematoxylin and eosin (H and E). (**A**) The control and (**C**) empagliflozin (EMPA)-treated groups show thin-walled alveoli (red arrows) and alveolar sacs (s). The thin-walled alveoli are made up of flat type I pneumocytes (black arrows) with densely pigmented nuclei and type II pneumocytes (dashed arrows) with big, rounded nuclei. (**B**) The bleomycin (BLM)-treated group reveals a disrupted architectural arrangement of the lung with markedly thickened alveolar septa (thick arrows) and extensive mononuclear cellular infiltration (curved arrows). Some alveoli appear collapsed (stars). Many of the alveolar epithelial lining cells exhibit vacuolar cytoplasmic necrosis with pyknotic nuclei (thin arrows). Severe interstitial hemorrhage (Hg) can be observed. (**D**) The EMPA+BLM-treated group shows many thin-walled alveoli (arrows) and alveolar sacs (S) with normal histology. However, there are still a few collapsed alveoli, minor alveolar wall thickening (thick arrows), and mild interstitial hemorrhage (Hg). Magnification is ×400; scale bar = 50 μm. (**E**) The histopathological and (**F**) fibrosis scoring among various studied groups (n = 6 rats in each group). * indicates a *p* < 0.05, ** indicates a *p* < 0.01, and *** indicates a *p* < 0.001.

**Figure 2 ijms-24-09481-f002:**
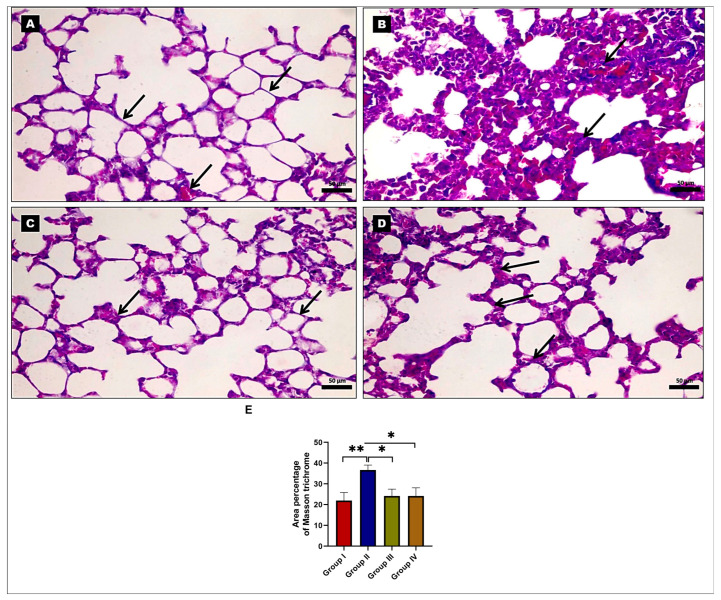
Representative photomicrographs of rat lung tissues stained with Masson’s trichrome. (**A**,**C**) The control and empagliflozin (EMPA)-treated groups showed little collagen fibers in the alveolar walls and around the blood vessels (arrows). (**B**) The bleomycin (BLM)-treated group showed massive deposits of collagen fibers (arrows). (**D**) The EMPA+BLM-treated group demonstrated alveolar walls with minimal collagen fiber deposition (arrows) (×400; scale bar = 50 μm). (**E**) A morphometric analysis of the mean area percentage in the Masson trichrome-stained lung tissue sections from various studied groups (*n* = 6 rats in each group). * indicates a *p* < 0.05 and ** indicates a *p* < 0.01.

**Figure 3 ijms-24-09481-f003:**
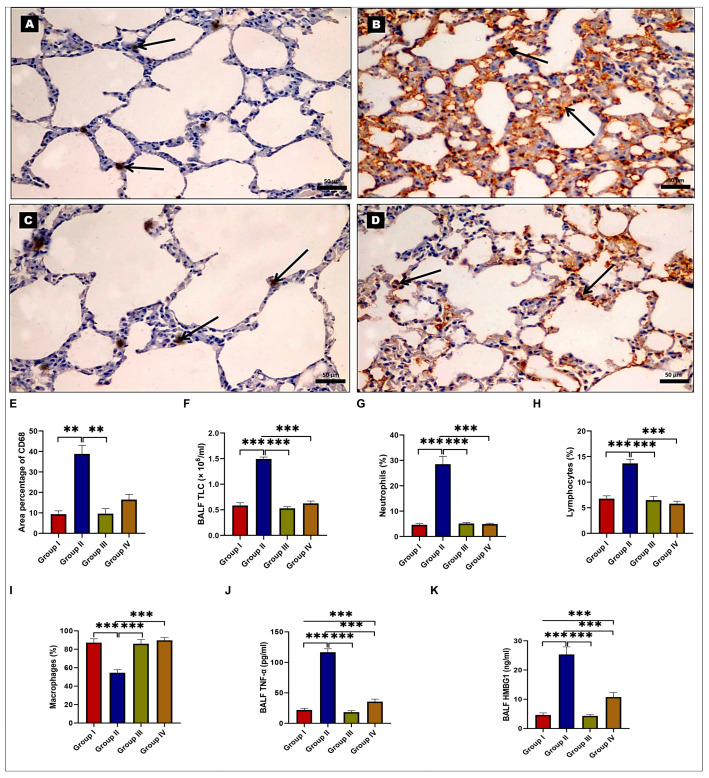
Effects of empagliflozin (EMPA) treatment on lung inflammatory status in a bleomycin (BLM)-induced pulmonary fibrosis model. Representative photomicrographs of CD68 immunohistochemical-stained lung sections. (**A**,**C**) The control and EMPA-treated groups show only a few positively stained macrophages (arrows) in the lung interalveolar septa. (**B**) The BLM-treated group exhibits an abundance of positively stained brown macrophages (arrows) in the interalveolar septa. (**D**) The EMPA + BLM-treated group shows a few positively stained brown macrophages (arrows) that are close to the control (×400, scale bar = 50 μm). (**E**) A morphometric analysis of the mean area percentage of CD68 immune reactivity; (**F**) the total leucocytic count; (**G**) neutrophil percentages; (**H**) lymphocyte percentages; (**I**) macrophage percentages; (**J**) bronchoalveolar lavage fluid (BALF) tumor necrosis factor-alpha (TNF-α); and (**K**) BALF high mobility group box 1 (HMBG1) among various studied groups (*n* = 6 rats in each group). Data are presented as mean and standard deviation (error bars) and analyzed using one-way analysis of variance (ANOVA) with Tukey’s post hoc test for multiple comparisons. ** indicates a *p* < 0.01 and *** indicates a *p* < 0.001.

**Figure 4 ijms-24-09481-f004:**
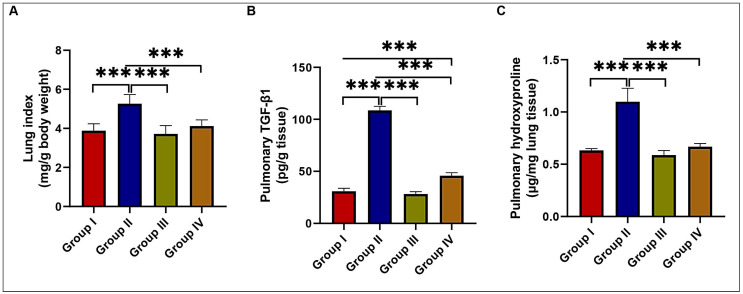
Effects of empagliflozin treatment on lung fibrotic states in a bleomycin-induced pulmonary fibrosis model. (**A**) The lung index; (**B**) pulmonary transforming growth factor beta 1 (TGF-β1); and (**C**) pulmonary hydroxyproline levels in various studied groups (*n* = 6 rats in each group). Data are presented as mean and standard deviation (error bars) and analyzed using one-way analysis of variance (ANOVA) with Tukey’s post hoc test for multiple comparisons. *** indicates a *p* < 0.001.

**Figure 5 ijms-24-09481-f005:**
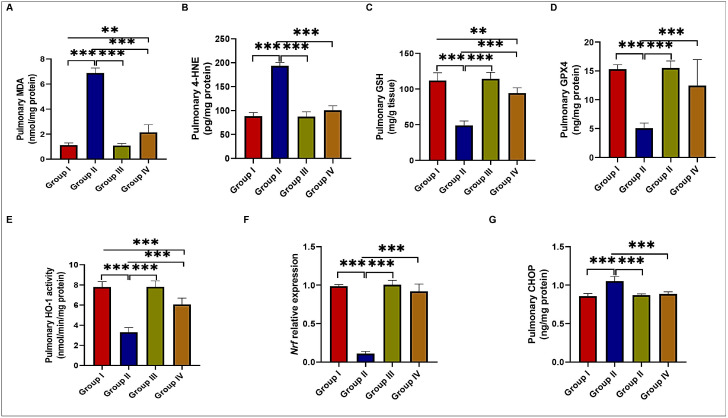
Effect of empagliflozin treatment on redox status, ferroptosis, and endoplasmic reticulum stress in a bleomycin-induced pulmonary fibrosis model. (**A**) Pulmonary malondialdehyde (MDA); (**B**) 4-hydroxynonenal (4-HNE); (**C**) reduced glutathione (GSH); (**D**) glutathione peroxidase 4 (GPX4); (**E**) heme oxygenase 1 (HO-1); (**F**) *Nrf2* relative expression; and (**G**) C/EBP homologous protein (CHOP) levels in various studied groups (*n* = 6 rats in each group). Data are presented as mean and standard deviation (error bars) and analyzed using one-way analysis of variance (ANOVA) with Tukey’s post hoc test for multiple comparisons. ** indicates a *p* < 0.01 and *** indicates a *p* < 0.001.

**Figure 6 ijms-24-09481-f006:**
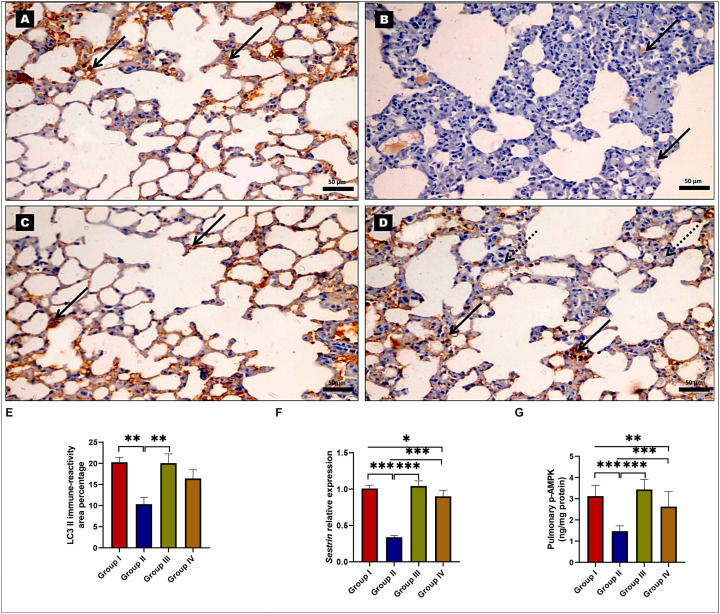
Effect of empagliflozin (EMPA) treatment on autophagy in a bleomycin (BLM)-induced pulmonary fibrosis model. Representative photomicrographs of LC3 II immunohistochemical-stained lung tissue sections showing (**A**,**C**) numerous cells with strong positive immunoreaction (arrows) in the form of brown cytoplasmic coloration in the control and EMPA-treated groups; (**B**) faint LC3 II immunoreaction (arrows) in the BLM-treated group; (**D**) a very strong LC3 II immunoreaction in some cells (arrows) with a weak LC3 II immunoreaction in few cells (dashed arrows) in the EMPA+BLM-treated group (×400, scale bar = 50 μm). (**E**) A morphometric analysis of the mean area percentage of LC3 II immune reactivity; (**F**) pulmonary *Sesn2* relative expression levels; and (**G**) pulmonary adenosine monophosphate-activated protein kinase (AMPK) levels among various studied groups (*n* = 6 rats in each group). Data are presented as mean and standard deviation (error bars) and analyzed using one-way analysis of variance (ANOVA) with Tukey’s post hoc test for multiple comparisons. * indicates a *p* < 0.05, ** indicates a *p* < 0.01, and *** indicates a *p* < 0.001.

**Figure 7 ijms-24-09481-f007:**
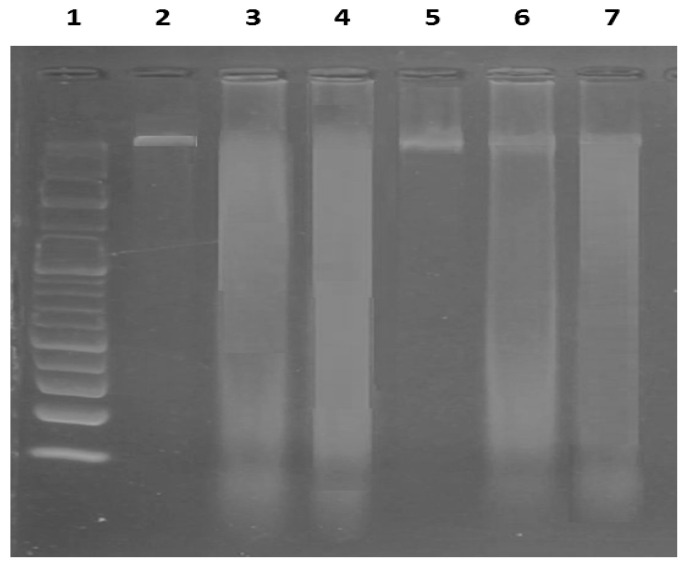
Electrophoretic pattern of DNA fragments. Lane 1: DNA marker (100–3000 bp); Lanes 2: the control group; Lanes 3 and 4: the bleomycin (BLM) group; Lanes 5: the empagliflozin (EMPA) group; and Lanes 6 and 7: the EMPA + BLM-treated group (*n* = 6 rats in each group).

**Figure 8 ijms-24-09481-f008:**
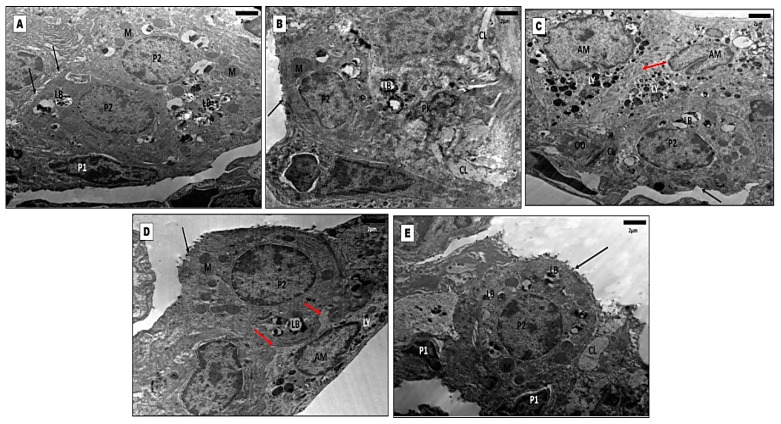
Representative electron micrographs from lung sections of the studied groups. (**A**) The control group shows type I pneumocytes (P1) with elongated euchromatic nuclei surrounded by a thin rim of cytoplasm. Type II pneumocytes (P2) with rounded euchromatic nuclei and intact microvillous borders (arrows). Their cytoplasm contains multiple mitochondria (M) and characteristic lamellar bodies (LB) with evident lamellae. A marked deposition of collagen fibers (CL) is seen in the interstitial tissue. (**B**) The bleomycin (BLM)-treated group shows type II pneumocytes (P2) with irregularly shrunken nuclei (PK), disturbed mitochondria (M), empty lamellar bodies (LB), and detached microvilli (arrows). (**C**) The BLM-treated group shows many alveolar macrophages (AM) with characteristic indented nuclei, many lysosomes (LY), and pseudopodia (red arrow). In the interstitial tissues, there is a deposition of collagen fibrils (CL) and oedematous fluid (OD). Type II pneumocytes (P2) depict empty lamellar bodies (LB) and the loss of microvilli (black arrow). (**D**) The empagliflozin (EMPA)-treated group shows normal type II pneumocytes (P2) with distinguishing cytoplasmic lamellar bodies (LB), mitochondria (M), and intact microvilli (black arrow). Alveolar macrophages (AM) with lysosomes (LY) and pseudopodia (red arrow) could be seen. (**E**) The EMPA + BLM-treated group displays type II pneumocytes (P2) with numerous lamellar bodies (LB) filled with surfactant, intact microvilli (arrow), many mitochondria (M), and big euchromatic nuclei with visible nucleoli. Parts of typical type I pneumocytes (P1) are seen. However, there are few interstitial collagen fibers (CL) visible (magnification ×1500, scale bar = 2 μm).

**Figure 9 ijms-24-09481-f009:**
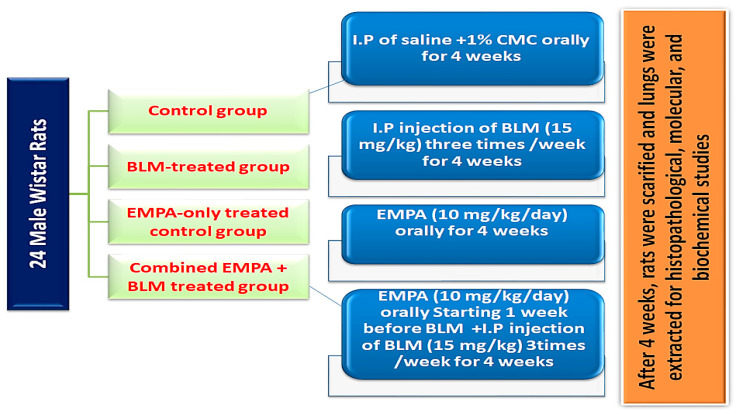
Schematic presentation of the experimental design.

## Data Availability

The data used in this study are available from the corresponding author upon request.

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
