# Peer review of "Empagliflozin Ameliorates Bleomycin-Induced Pulmonary Fibrosis in Rats by Modulating Sesn2/AMPK/Nrf2 Signaling and Targeting Ferroptosis and Autophagy"

_ijms, 2023, doi:10.3390/ijms24119481_

Round 1

Reviewer 1 Report

Hemat El-Sayed El-Horany et al. describes the effect of empagliflozin in the bleomycin-induced pulmonary fibrosis model in rats that regulate the effect on Sesn2/AMPK/Nrf2/HO-1 signaling pathway, as well as its modulatory role in ferroptosis, inflammation, ER stress, and autophagy.

The article is very well written and very well addressed, however, I have a couple of minor observations.

Results

The results are presented in a very appropriate manner

Methodology

Figure 9 is missing

For better reproducibility, add a brief description of the Hydroxyproline method and please set the concentration of the antibodies.

Conclusions:

The authors state, "These findings suggest that SGLT2 inhibitor therapy, such as EMPA, could be a potential protective approach for pulmonary complications in patients receiving BLM”, they cannot argue with these results because they do not measure it. However, it would be very interesting, bleomycin might even lose its "therapeutic" effect in such persons.

The references are appropriate

Author Response

Dear Reviewer 1

Many thanks for your constructive and valuable criticisms. Our responses are presented below and we are looking forward and ready to respond to any future comment (s)

Reviewer 1:

Hemat El-Sayed El-Horany et al. describes the effect of empagliflozin in the bleomycin-induced pulmonary fibrosis model in rats that regulate the effect on Sesn2/AMPK/Nrf2/HO-1 signaling pathway, as well as its modulatory role in ferroptosis, inflammation, ER stress, and autophagy.

  • The article is very well written and very well addressed, however, I have a couple of minor observations.

Response: Thank you very much for recommending our manuscript and for your encouraging comments and we are pleased to respond to them. 

  • Results
  • The results are presented in a very appropriate manner

Response: Thank you very much for recommending our manuscript and for your encouraging comments.

  • Methodology
  • Figure 9 is missing

Response: Thank you for your careful review. We felt sorry for this mistake. Figure 9 has been added in its right place in the manuscript.

  • For better reproducibility, add a brief description of the Hydroxyproline method and please set the concentration of the antibodies.

Response: Thank you for your insightful comments. We have carefully considered your valuable suggestions and included a brief description of the hydroxyproline method as follows:

In brief, samples were hydrolyzed with 6 N HCl for 24 hours at 120 °C, then treated with the buffer and chloramine T reagent for 20 minutes at room temperature before being treated with perchloric acid for 15 minutes at 60 °C. After cooling, the absorbance of the red complex was measured at 550 nm.”

In addition, the concentration of the antibodies has been specified according to your professional recommendations as follows:

The sections were then incubated at 4 °C with the primary antibodies, rabbit monoclonal anti-LC3II (0.5 mg/mL; 1:400; catalog no. ab232940; Abcam, Massachusetts, USA) and macrophage CD68 (1 mg/mL; 1:400; Bio-Rad, Alfred Nobel Drive, Hercules, California, USA).”

  • Conclusions:

The authors state, "These findings suggest that SGLT2 inhibitor therapy, such as EMPA, could be a potential protective approach for pulmonary complications in patients receiving BLM”, they cannot argue with these results because they do not measure it. However, it would be very interesting, bleomycin might even lose its "therapeutic" effect in such persons.

Response: Thank you for your insightful comments. We totally agree with your point of view. Accordingly, we omitted this statement and included recommendations for further studies to confirm the protective effect of SGLT2 inhibitor therapy, such as EMPA, against pulmonary complications in patients receiving BLM and to determine the extent to which it can antagonize BLM's therapeutic effect in such patients by promoting chemoresistance. We included the following statement to the conclusion:

However, further studies are warranted to confirm the protective effect of SGLT2 inhibitor therapy, such as EMPA, against pulmonary complications in patients receiving BLM and to determine the extent to which it can antagonize BLM's therapeutic effect in such patients by promoting chemoresistance.” 

  • The references are appropriate

Response: Thank you very much for recommending our manuscript and for your encouraging comments.

Reviewer 2 Report

the manuscript is interesting and generally well written. However, it presents some flaws that must be resolved. In particular:

Lines 73-77:  the role of Nrf2 signalling is poorly introduced. In fact, it deserves to be highlighted that NRF2 plays an important role in cancer prevention and progression (PMID: 36641100, 36335520, 36289931 ). This is an important point to add since it can further highlight the interesting results obtained by the authors.

Figure 1 and 2: The H&E images must be improved. In particular, the magnifications are too low to appreciate the morphology of the tissue and the localization of the cells showed by the arrows. 

Figure 3 and 6: the quality of the images is very low. Higher magnifications are needed

4.1. Chemicals: Product codes must be reported

Figure 9 is not shown

Authors must report the number of rats analysed in the legend of each figure

An accurate revision of typing errors is recommended

Author Response

Dear Reviewer 2

Many thanks for your constructive and valuable criticisms. Our responses are presented below and we are looking forward and ready to respond to any future comment (s)

Reviewer 2:

  • the manuscript is interesting and generally well written. However, it presents some flaws that must be resolved. In particular:

Response: Thank you very much for recommending our manuscript and for your encouraging comments and we are pleased to respond to them. 

  • Lines 73-77: the role of Nrf2 signalling is poorly introduced. In fact, it deserves to be highlighted that NRF2 plays an important role in cancer prevention and progression (PMID: 36641100, 36335520, 36289931 ). This is an important point to add since it can further highlight the interesting results obtained by the authors.

Response: Thank you for your insightful comment. We thoroughly studied these interesting studies. According to the suggested studies, Nrf2 plays an important bidirectional role in cancer prevention and progression. In response to oxidative and carcinogenic stimuli, Nrf2 activates various genes involved in defensive and adaptive pathways to prevent normal tissue damage. Therefore, Nrf2 activation executes a protective role under physiological conditions. However, it promotes cancer development, metastasis, and drug resistance after cancer is established. Selective Nrf2 modulators can be used as adjuvant therapy after conventional chemotherapy, targeted therapy, and immunotherapy.

We defined these concepts in the introduction following the recommended studies, and we appreciate your suggestions, which helped us improve our manuscript.

Nuclear factor erythroid 2-related factor 2 (Nrf2) plays an important bidirectional role in cancer prevention and progression. As a well-known cytoprotective factor, it plays an important role in protecting cells from oxidative damage caused by increased reactive oxygen species (ROS) levels. When activated by ROS, Nrf2 induces the expression of antioxidant enzymes by binding to the promoters of target genes containing antioxidant response elements, thereby contributing to the synthesis and renewal of glutathione (GSH), a major GPX4 cofactor, and detoxifying molecules [10, 11]. Therefore, Nrf2 activation plays a protective role under physiological conditions. However, Nrf2 activation in cancer cells promotes cancer progression and metastasis, as well as chemoresistance by inactivating drug-mediated oxidative stress and protecting cancer cells from drug-induced cell death. Selective Nrf2 modulators can be used as adjuvant therapy after conventional chemotherapy, targeted therapy, and immunotherapy [11]. Targeting the Nrf2 pathway in cancer cells could pave the way for the development of more effective anticancer drugs or improve the efficacy of existing drugs by reducing drug resistance [12]. Furthermore, Nrf2 expression can be reactivated by natural or synthetic compounds via downregulating histone deacetylases and DNA methyltransferases, inhibiting Nrf2 promoter methylation and protecting normal cells from ROS damage and tumorigenesis [13].

References:

  1. Tossetta, G.; Marzioni, D., Targeting the NRF2/KEAP1 pathway in cervical and endometrial cancers. European journal of pharmacology 2023, 941, 175503.
  2. Ghareghomi, S.; Habibi-Rezaei, M.; Arese, M.; Saso, L.; Moosavi-Movahedi, A. A., Nrf2 Modulation in Breast Cancer. Biomedicines 2022, 10, (10).
  3. Marzioni, D.; Mazzucchelli, R.; Fantone, S.; Tossetta, G., NRF2 modulation in TRAMP mice: an in vivo model of prostate cancer. Molecular biology reports 2023, 50, (1), 873-881
  • Figure 1 and 2: The H&E images must be improved. In particular, the magnifications are too low to appreciate the morphology of the tissue and the localization of the cells showed by the arrows.

Response: Thank you for your insightful comments. We replaced the images with new ones with higher magnification to improve the quality of images. We also indicated the cell localization by arrows.

  • Figure 3 and 6: the quality of the images is very low. Higher magnifications are needed

Response: Thank you for your insightful comments. We replaced the images with new ones with higher magnification to improve the quality of images.

  • 1. Chemicals: Product codes must be reported

Response: Thank you for your insightful comments. We have specified the product codes according to your professional recommendations as follows:

Bleomycin (15 U/vial) was purchased from Celon Laboratories Pvt. Ltd, India (NDC 61703-332-18). Empagliflozin (Jardiance) was obtained from Boehringer Ingelheim, Germany (NDC 0597-0152-30)”

  • Figure 9 is not shown

Response: Thank you for your careful review. We felt sorry for this mistake. Figure 9 has been added in its right place in the manuscript.

  • Authors must report the number of rats analysed in the legend of each figure

Response: Thank you for your insightful comments. The number of rats has been included in the legend of each figure.

  • An accurate revision of typing errors is recommended

Response: Thank you for your valuable comment. The whole manuscript has been accurately revised for typing mistakes.

Round 2

Reviewer 2 Report

the manuscript has been significantly improved and can be accepted in the present form